# Are CT-Derived Muscle Measurements Prognostic, Independent of Systemic Inflammation, in Good Performance Status Patients with Advanced Cancer?

**DOI:** 10.3390/cancers15133497

**Published:** 2023-07-05

**Authors:** Josh McGovern, Ross D. Dolan, Claribel Simmons, Louise E. Daly, Aoife M. Ryan, Derek G. Power, Marie T. Fallon, Barry J. Laird, Donald C. McMillan

**Affiliations:** 1Academic Unit of Surgery, New Lister Building, Royal Infirmary, Glasgow G31 2ER, UK; ross.dolan@glasgow.ac.uk (R.D.D.);; 2Institute of Genetics and Molecular Medicine, University of Edinburgh, Edinburgh EH4 2XU, UK; claribel.simmons@nhs.scot (C.S.); barry.laird@ed.ac.uk (B.J.L.); 3Cork Cancer Research Centre, University College Cork, T12 YN60 Cork, Ireland; louise.daly@danone.com (L.E.D.); a.ryan@ucc.ie (A.M.R.); 4Department of Medical Oncology, Mercy and Cork University Hospital, T12 DC4A Cork, Ireland; dpower@muh.ie

**Keywords:** sarcopenia, inflammation, clinical outcomes, cancer

## Abstract

**Simple Summary:**

Cancer cachexia is associated with a loss of skeletal muscle. The analysis of CT images has facilitated the measurement of such changes. Whether these measurements have prognostic value for survival in patients with advanced cancer when adjusted for inflammation has recently been questioned. Therefore, the aim of the present study was to examine if CT-derived skeletal muscle measures were related to survival, independent of how inflamed the patient is. Our results suggest that CT-derived muscle measurements do not have independent prognostic value for survival in patients with advanced cancer when the inflammatory status of the patients is considered.

**Abstract:**

The present study examined the relationships between CT-derived muscle measurements, systemic inflammation, and survival in advanced cancer patients with good performance status (ECOG-PS 0/1). Data was collected prospectively from patients with advanced cancer undergoing anti-cancer therapy with palliative intent. The CT Sarcopenia score (CT-SS) was calculated by combining the CT-derived skeletal muscle index (SMI) and density (SMD). The systemic inflammatory status was determined using the modified Glasgow Prognostic Score (mGPS). The primary outcome of interest was overall survival (OS). Univariate and multivariate Cox regressions were used for survival analysis. Three hundred and seven patients met the inclusion criteria, out of which 62% (n = 109) were male and 47% (n = 144) were ≥65 years of age, while 38% (n = 118) were CT-SS ≥ 1 and 47% (n = 112) of patients with pre-study blood were inflamed (mGPS ≥ 1). The median survival from entry to the study was 11.1 months (1–68.1). On univariate analysis, cancer type (*p* < 0.05) and mGPS (*p* < 0.001) were significantly associated with OS. On multivariate analysis, only mGPS (*p* < 0.001) remained significantly associated with OS. In patients who were ECOG-PS 0, mGPS was significantly associated with CT-SS (*p* < 0.05). mGPS may dominate the prognostic value of CT-derived sarcopenia in good-performance-status patients with advanced cancer.

## 1. Introduction

Contemporary evidence suggests there are around 167,000 cancer deaths in the UK every year [1]. Furthermore, nearly half of all newly diagnosed cancer cases were locally advanced or metastatic, where treatment options are limited [1]. Given that most patients with advanced disease are likely to die from their malignancy, there is continued interest in identifying prognostic factors for survival in patients with advanced cancer, particularly factors independent of performance status [2].

CT-derived muscle measures, skeletal muscle index (SMI) and density (SMD), have been reported to have a detrimental impact on survival outcomes [3]. However, a recent systematic review reported that low SMI and SMD had similar prevalence across cancer types and disease stages [4]. As such, it was hypothesized that body composition alone was not the main determinant of survival. Furthermore, it is imperative that such measures be used in conjunction with other factors, such as performance status and systemic inflammation [5].

This issue was highlighted in a recent study by Hacker and co-workers, who reported in a study of advanced oesophagogastric patients with good performance status (ECOG-PS 0/1), that although associated with ECOG-PS and systemic inflammation (modified Glasgow prognostic score (mGPS)), CT-derived muscle measures were not independently associated with survival [6]. Furthermore, they suggested that cancer-related systemic inflammation represented the main causal association with poorer survival.

If Hacker’s observations were confirmed in future studies, then it would have implications for the utility of CT-derived muscle measures as biomarkers in clinical practice. Specifically, whether such body composition measures add prognostic information to the recognized framework of ECOG-PS and mGPS in patients with advanced cancer [7]. Furthermore, to their use as surrogate markers of nutritional status for the diagnosis of cancer cachexia [8]. Therefore, the aim of the present study was to examine the relationships between CT-derived muscle measurements, systemic inflammation, and survival in good-performance status patients with advanced lung and gastrointestinal cancer.

## 2. Materials and Methods

### 2.1. Patients

Patients with advanced cancer were identified from an international database. Data were prospectively collected across nine sites in the UK and Ireland between 2011 and 2016 and retrospectively analyzed [9,10,11]. Patients with either advanced lung or GI cancer (defined as locally advanced or with evidence of metastasis) who had good performance stats (ECOG-PS 0/1) and suitable pre-treatment CT images for body composition analysis were considered for inclusion. The primary endpoint was overall survival from entry to the study. The date of death was confirmed using hospital electronic records until 18 June 2018, which served as the censor date.

Clinicopathological characteristics were recorded for each patient within the 3 months prior to study entry. Primary cancer types were broadly classified as GI, lung, or other. The presence of metastatic disease was identified from staging CT imaging obtained prior to study entry. BMI was categorized as <25/≥25 kg/m^2^. ECOG-PS was assessed by a clinician or clinical researcher at the institute where the patient was receiving treatment at entry to the study and categorized as 0 or 1. The mGPS was determined from venous blood samples obtained at study entry. The mGPS was calculated as previously described (CRP ≤ 10 mg/L = 0, CRP > 10 mg/L, and albumin ≥ 35 g/L = 1, CRP > 10 mg/L, and albumin < 35 g/L = 2) [12]. Serum CRP (mg/L) and albumin (g/L) concentrations were measured using an autoanalyzer.

### 2.2. CT-Body Composition Analysis

CT images, obtained at the level of the third lumbar vertebra, were analyzed as previously described [13]. Patients whose scans were taken three months or more prior to commencing anti-cancer treatment and had significant movement artifacts or missing regions of interest were excluded. All images were analyzed using the free-ware program Image J, version 1.47, (http://rsbweb.nih.gov/ij/ (accessed on 1 June 2022)).

The skeletal muscle area (SMA, cm^2^) was measured by manually delineating muscle areas of the quadratus lumborum, psoas, rectus abdominus, erector spinae muscles, internal transverse, and external oblique muscle groups. Attenuation thresholds for muscle were −29 to +150 HU. SMA measurements were then normalized by division of the patient’s height in meters squared to generate the skeletal muscle index (SMI, cm^2^/m^2^). Skeletal muscle radiodensity (SMD, HU) was measured from the same ROI used to calculate SMI as its mean HU.

These indices were then compared with established thresholds for body composition status [14]. Specifically, a low SMI was defined as an SMI < 43 cm^2^/m^2^ if BMI < 25 kg/m^2^ and an SMI < 53 cm^2^/m^2^ if BMI ≥ 25 kg/m^2^ in male patients and an SMI < 41 cm^2^/m^2^ in female patients if BMI < or ≥ 25 kg/m^2^ [14]. A low SMD was defined as an SMD < 41 HU in patients with BMI < 25 kg/m^2^ and < 33 HU in patients with BMI > 25 kg/m^2^ [14].

SMI and SMD measurements were combined to form the CT-Sarcopenia score (CT-SS), shown to have complementary prognostic value for survival in patients with cancer [15,16]. Patients were categorized as normal/high SMI and normal/high SMD = CT-SS 0, low SMI and normal/high SMD = CT-SS 1, and low SMI and low SMD = CT-SS 2, as previously described [16].

### 2.3. Statistical Analysis

Demographic data, clinicopathological variables, SMI, SMD, CT-SS, ECOG-PS, mGPS, and overall survival (OS) were presented as categorical variables. Categorical variables were analyzed using χ^2^ test for linear-by-linear association.

Univariate and multivariate survival data were analyzed using Cox’s proportional hazards model. Variables associated with OS at a significance level of *p* < 0.1 on univariate analysis were included in the multivariate model using a backward conditional approach. OS was defined as the time (months) from the entry to study to the date of death due to any cause.

Missing data were excluded from analysis on a variable-by-variable basis. Two-tailed *p* values < 0.05 were considered statistically significant. Statistical analysis was performed using SPSS software version 28.0 (SPSS Inc., Chicago, IL, USA).

## 3. Results

Three hundred and seven patients met the inclusion criteria (see Figure 1). The clinicopathological characteristics of the patients included in the study are shown in Table 1. Out of 307 patients, 62% (n = 109) were male and 47% (n = 144) were ≥65 years of age; 68% (n = 208) had GI tumors and 32% (n = 99) had lung tumors; 87% (n = 268) had metastatic disease on staging CT imaging; 92% (n = 283) received chemotherapy prior to study entry and 6% (n = 19) received radiotherapy; 51% (n = 155) were overweight (BMI ≥ 25 kg/m^2^); 38% (n = 118) had a low SMI; and 48% (n = 142) had a low SMD. Sixty-two percent (n = 189) of patients were CT-SS 0, 16% (n = 48) were CT-SS 1, and 23% (n = 70) were CT-SS 2; 48% (n = 146) of patients were ECOG-PS 0 and 52% (n = 161) were ECOG-PS 1; 47% (n = 112) of patients with blood facilitating calculation of mGPS (n = 240) were inflamed (mGPS ≥ 1). From entry to the study, the median survival of patients was 11.1 months (1–68.1).

**Table 1 cancers-15-03497-t001:** The relationship between the CT-SS and clinicopathological characteristics, ECOG-PS, systemic inflammation, and overall survival in patients with advanced cancer who had good performance status (ECOG-PS 0/1), stratified by the CT-SS (n = 307).

	CT-SS 0	CT-SS 1	CT-SS 2	*p* Value ^1^
	(n = 189)	(n = 48)	(n = 70)	
Age				0.042
<65	105 (56%)	32 (67%)	26 (37%)	
65–74	56 (30%)	9 (19%)	29 (42%)	
>74	28 (15%)	7 (15%)	15 (21%)	
Sex				<0.001
Female	55 (29%)	23 (48%)	39 (56%)	
Male	134 (71%)	25 (52%)	31 (44%)	
Cancer Type				0.431
Lung	60 (32%)	12 (25%)	27 (38%)	
GI	129 (68%)	36 (75%)	43 (61%)	
Metastatic disease				0.157
No	29 (15%)	3 (6%)	7 (10%)	
Yes	160 (85%)	45 (94%)	63 (90%)	
Chemotherapy				0.859
Yes	174 (92%)	44 (92%)	65 (93%)	
No	15 (8%)	4 (8%)	5 (7%)	
Radiotherapy				0.339
Yes	10 (5%)	3 (6%)	6 (9%)	
No	179 (95%)	45 (94%)	64 (91%)	
BMI (kg/m^2^)				0.014
<25	86 (46%)	21 (44%)	45 (64%)	
≥25	103 (54%)	27 (56%)	25 (36%)	
Low SMI				<0.001
No	189 (100%)	0 (0%)	0 (0%)	
Yes	0 (0%)	48 (100%)	70 (100%)	
Low SMD				<0.001
No	109 (60%)	46 (100%)	0 (0%)	
Yes	72 (40%)	0 (0%)	70 (100%)	
ECOG-PS				0.197
0	93 (49%)	26 (54%)	27 (39%)	
1	96 (51%)	22 (46%)	43 (61%)	
mGPS ^2^				0.058
0	78 (55%)	24 (62%)	26 (44%)	
1	23 (16%)	3 (8%)	5 (8%)	
2	41 (29%)	12 (31%)	28 (48%)	
Overall survival				0.548
Yes	43 (23%)	12 (25%)	13 (19%)	
No	146 (77%)	36 (75%)	57 (81%)	

ECOG-ps—Eastern Cooperative Oncology Group Performance Status; mGPS—modified Glasgow Prognostic Score. ^1^
*p* value is from χ^2^ analysis. ^2^ 67 patients did not have blood at study entry to calculate mGP. The relationship between OS and clinicopathological characteristics, CT-derived muscle measurements, ECOG-PS, and systemic inflammation in patients with advanced cancer is shown in Table 2. On univariate analysis, cancer type (*p* < 0.05) and mGPS (*p* < 0.001) were significantly associated with OS. On univariate analysis, neither age (*p* = 0.146), sex (*p* = 0.691), metastatic disease (*p* = 0.995), chemotherapy (*p* = 0.606), radiotherapy (*p* = 0.112), BMI (*p* = 0.805), CT-SS (*p* = 0.421), nor ECOG-PS (*p* = 0.142) were significantly associated with OS. On multivariate analysis, only mGPS (*p* < 0.001) remained significantly associated with OS.

The relationship between the CT-SS and clinicopathological characteristics, CT-derived muscle measurements, ECOG-PS, systemic inflammation, and overall survival in patients with advanced cancer is shown in Table 1. On univariate analysis, the CT-SS was significantly associated with age (*p* < 0.05), sex (*p* < 0.001), BMI (*p* < 0.05), low SMI (*p* < 0.001), and low SMD (*p* < 0.001). On univariate analysis, the CT-SS was not significantly associated with cancer type (*p* = 0.431), metastatic disease (*p* = 0.157), chemotherapy (*p* = 0.859), radiotherapy (*p* = 0.339), ECOG-PS (*p* = 0.197), mGPS (*p* = 0.058), or OS (*p* = 0.548).

The relationship between ECOG-PS, mGPS, and CT-SS in patients with advanced cancer is shown in Table 3. In patients who were ECOG-PS 0, mGPS was significantly associated with CT-SS (*p* < 0.05). In patients who were ECOG-PS 1, mGPS was not significantly associated with CT-SS (*p* = 0.602). In patients with mGPS 0, ECOG-PS was not significantly associated with CT-SS (*p* = 0.286). In patients who were mGPS 1, ECOG-PS was not significantly associated with CT-SS (*p* = 0.739). In patients who were mGPS 2, ECOG-PS was not significantly associated with CT-SS (*p* = 0.251).

## 4. Discussion

The results of the present study confirmed that mGPS was independently associated with survival in patients with good performance status with advanced lung and GI cancer. However, the independent prognostic value of CT-derived sarcopenia (CT-SS) was not confirmed and would suggest that, given that systemic inflammation and sarcopenia are closely associated, systemic inflammation is the dominant route by which survival is compromised in these patients. The present observations therefore support those of Hacker and co-workers, who recently questioned the independent prognostic value of CT-derived muscle measures in patients with advanced cancer [6]. Moreover, they support the concept that systemic inflammation (mGPS) dominates the prognostic value of CT-derived sarcopenia in good performance status patients with advanced cancer. Taken together, the results may have implications for the current framework proposed by the Global Leadership in Malnutrition (GLIM) for diagnosing cancer cachexia. Specifically, whether phenotypic or etiologic criteria are more important prognostic factors in good performance status patients [17].

The CT-sarcopenia score (CT-SS), considered to objectively characterize the nutritional and functional reserve of the cancer patient, is thought to have complementary prognostic value for survival [16]. Indeed, the CT-SS was reported to be significantly associated with survival in a study of 1002 patients undergoing potentially curative surgery for primary colorectal cancer [16]. Similar results were seen in a study of 232 patients with oesophagogastric cancer undergoing neoadjuvant chemotherapy (NAC) [15]. In contrast, the CT-SS was not significantly associated with survival in the present study. While heterogeneity exists between the studies, specifically the stage of disease, it was of interest that a similar prevalence of CT-SS ≥ 1 was reported in those with primary colorectal cancer (51%), primary oesophagogastric cancer (36%), and those with locally advanced/metastatic cancer in the present study (39%). However, nearly twice as many patients were considered to be inflamed (mGPS ≥ 1) in the present cohort (47%), compared to patients with primary colorectal (27%) and oesophagogastric cancer (28%). As such, the present observations are in keeping with those of a review by McGovern and co-workers, who reported that given a low SMI and SMD prevalence across cancer types and disease stages, CT-derived muscle measures are not the main determinant of survival [4]. Furthermore, support the hypothesis that CT-derived body composition measures should be used in conjunction with other factors, such as systemic inflammation [4].

When the present results are compared with those of Hacker and co-workers [6], although differences exist in the tumor subtypes and sample size, the distribution of ECOG-PS (ECOG-PS 0 = 48% vs. 57%) and inflammatory status (mGPS ≥ 1= 47% vs. 49%) of included patients was similar. In contrast, there were significant differences in the median SMI and SMD reported (median SMI 47.0 cm^2^/m^2^ vs. 61.6 cm^2^/m^2^ and median SMD 38 HU vs. 46.2 HU, respectively). Given that age, gender, and BMI are all likely to be confounding factors in CT-derived muscle analysis, the contrasting observations may be explained by an increased prevalence of female patients (38% vs. 24%), older patients (47% ≥ 65 years of age vs. 26%), and obese patients (51% BMI ≥ 25 kg/m^2^ vs. 33%) in the present study, compared with that of Hacker and co-workers. Furthermore, significant differences in muscle status have been noted when comparing studies of CT-derived muscle measures from different European countries, highlighting that lifestyle and diet may also be confounding factors [5]. As such, while the present observations support those of Hacker and co-workers, given their clinical significance, further study is warranted to examine whether mGPS dominates the prognostic value of CT-derived sarcopenia in advanced cancer patients with good performance status.

The assessment of CT images acquired as part of standard cancer care has demonstrated that cancer cachexia is associated with a loss of skeletal muscle mass (low SMI) and reduced muscle radiation attenuation (low SMD) [18]. Despite being considered objective surrogate markers of nutritional status, Arulananda and Segelov recently questioned the clinical utility of CT-derived muscle measures given their relative prognostic value [19]. If the present observations are confirmed in future studies, then they may have implications for both the diagnosis and management of cancer cachexia. Specifically, the currently proposed GLIM criteria include both reduced muscle mass, for which SMI has been shown to be a reliable measure [20], and inflammation [17]. However, if inflammation dominates the prognostic value of SMI, then consideration should be given to whether it becomes the dominant criterion for identifying cachexia in patients with cancer [6,21,22,23,24]. Therefore, the present results support the hypothesis that cancer cachexia should be considered “disease-related inflammation with malnutrition” rather than the current consensus of “disease-related malnutrition with inflammation [8].

ECOG-PS remains an important determinant of eligibility for anti-cancer treatment [7], with almost all good performance status patients conventionally considered candidates for optimal treatment. Furthermore, ECOG-PS is universally utilized as a tool for stratifying eligibility for randomized clinical trials, with only patients with good performance status (ECOG-PS 0/1) generally considered suitable [25]. However, the subjective nature of performance status has implications for the external validity of clinical trials in real-world clinical practice [26]. Indeed, contemporary evidence is challenging the exclusion of ECOG-PS 2 patients from randomized clinical trials of immunotherapy, with recent studies by Yang and co-workers and Singh and co-workers reporting that the inclusion of ECOG-PS 2 patients did not adversely affect trial outcomes [25,27]. Furthermore, there is thought to be significant heterogeneity in ECOG-PS 2 patients, with continued interest in identifying additional predictive biomarkers that can further stratify likely outcomes in such patients [28]. Examples include biomarkers of the nutritional status of the patient, such as CT-derived muscle measures and systemic inflammation. However, while both factors have prognostic value for clinical outcomes in patients with advanced cancer, their close association questions the causality of these relationships and the order of dominance. The present observations, together with those in clinical trials [6,23,29], favor a framework where the systemic inflammatory response is the dominant factor to be used in patients with good performance status to predict the likely outcome.

There are several limitations to the present study. Principally, the analysis is retrospective on a prospective dataset and may be subject to sample bias. Secondly, in contrast to contemporary literature, CT-derived skeletal muscle measures were not independently associated with survival in the present study. This may be explained by the sample size or the inclusion of only patients with good performance status. Nevertheless, these measurements are available in routine clinical practice, and the observations should be readily tested. Lastly, the observations of this modest-sized study of good performance status patients with advanced lung and GI cancer suggest that mGPS may dominate the prognostic value of CT-derived muscle measures, in keeping with contemporary literature [7]. However, further large cohort studies across a range of tumor subtypes are required to determine the order of dominance in good-performance status patients with advanced cancer.

## 5. Conclusions

In keeping with contemporary literature, the results of the present study support the hypothesis that mGPS may dominate the prognostic value of CT-derived sarcopenia in good-performance status patients with advanced cancer. These results may have implications for how CT-derived SMI may be used in the currently proposed GLIM framework to diagnose cancer cachexia.

## Figures and Tables

**Figure 1 cancers-15-03497-f001:**
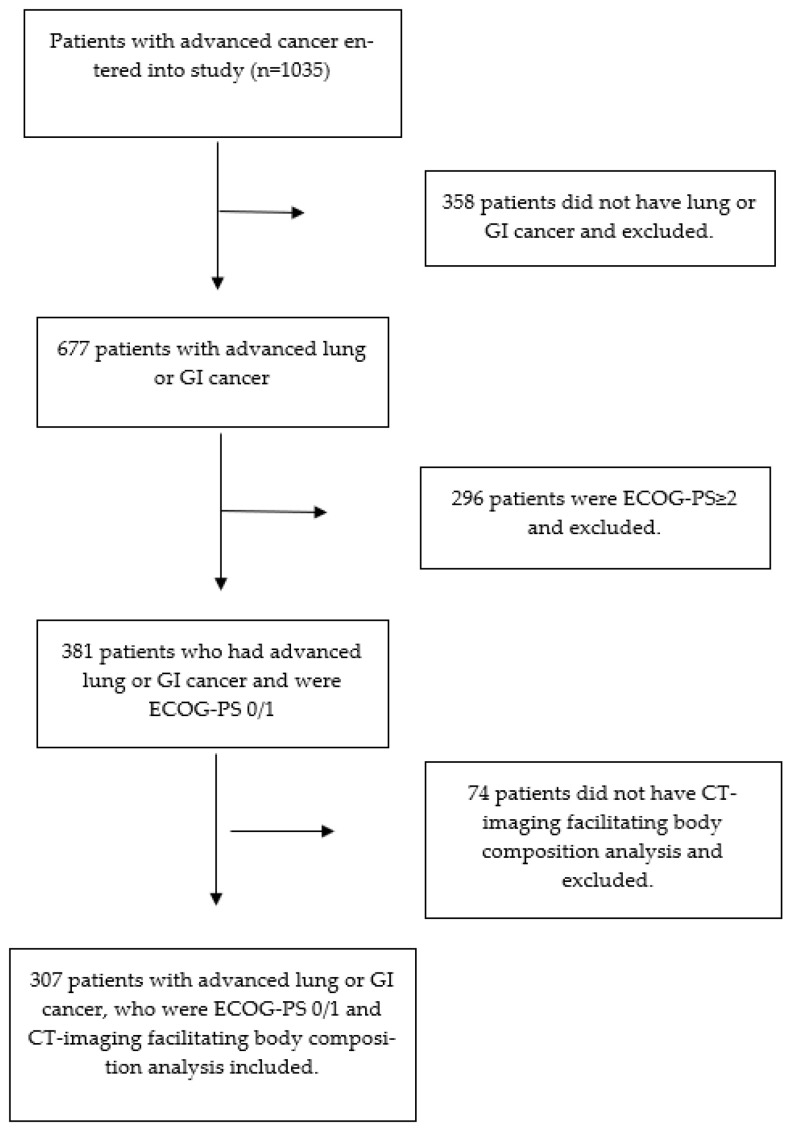
Flow diagram of the included patients.

**Table 2 cancers-15-03497-t002:** The relationship between overall survival and clinicopathological characteristics, CT-derived body composition measures, ECOG-PS, and systemic inflammation in patients with advanced cancer and good performance status (n = 307).

	Univariate Analysis		Multivariate Analysis	
	Hazard Ratio (95% Confidence Interval)	*p*-Value	Hazard Ratio (95% Confidence Interval)	*p*-Value
Age (<65/65–74/>74)	0.88 (0.74–1.05)	0.146	-	-
Sex (Female/Male)	0.95 (0.73–1.23)	0.691	-	-
Cancer type (Lung/GI)	0.66 (0.50–0.87)	0.003	-	0.119
Metastatic disease (No/Yes)	1.00 (0.69–1.46)	0.995	-	-
Chemotherapy (No/Yes)	0.87 (0.50–1.49)	0.606	-	-
Radiotherapy (No/Yes)	1.79 (0.87–3.68)	0.112	-	-
BMI (<25/≥25, kg/m^2^)	0.97 (0.75–1.25)	0.805	-	-
CT-SS (0/1/2)	1.06 (0.92–1.24)	0.421	-	-
ECOG-PS (0/1)	1.21 (0.94–1.56)	0.142	-	-
mGPS (0/1/2)	1.33 (1.13–1.55)	<0.001	1.33 (1.13–1.55)	0.001

**Table 3 cancers-15-03497-t003:** The relationship between ECOG-PS, mGPS, and CT-SS in patients with advanced cancer (n = 240).

	mGPS 0 (n = 128)	mGPS 1 (n = 31)	mGPS 2 (n = 81)	*p* Value ^1^
ECOG-PS 0 (n = 146)	CT-SS 0 37 (65%) CT-SS 1 11 (19%) CT-SS 2 9 (16%)	CT-SS 0 11 (73%) CT-SS 1 1 (7%) CT-SS 2 3 (20%)	CT-SS 0 11 (39%) CT-SS 1 6 (21%) CT-SS 2 11 (39%)	0.016
ECOG-PS 1 (n = 161)	CT-SS 0 41 (57%) CT-SS 1 13 (18%) CT-SS 2 17 (24%)	CT-SS 0 12 (76%) CT-SS 1 2 (12%) CT-SS 2 2 (12%)	CT-SS 0 30 (57%) CT-SS 1 6 (11%) CT-SS 2 17 (32%)	0.602
*p* value ^1^	0.286	0.739	0.251	

^1^ *p* value is from chi-square analysis.

## Data Availability

Raw data will be made available on request to the senior author (DCM).

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
