# Peer review of "Are CT-Derived Muscle Measurements Prognostic, Independent of Systemic Inflammation, in Good Performance Status Patients with Advanced Cancer?"

_cancers, 2023, doi:10.3390/cancers15133497_

Round 1
Reviewer 1 Report
In this short paper, the relationships between CT-derived muscle measurements, systemic inflammation, and survival in good performance status patients (ECOG-PS 0/1) with advanced cancer are investigated. CT-derived muscle measures, skeletal muscle index (SMI), and density (SMD) were combined to form the CT Sarcopenia score (CT-SS). For systemic inflammation was employed the modified Glasgow Prognostic Score (mGPS). Primary outcome of interest was overall survival (OS).
Survival analysis has been performed using univariate and multivariate Cox regression.
The results have shown that in patients with advanced lung or GI cancer, who were good performance status (ECOG-PS 0/1), mGPS was independently associated with survival. In contrast, although ECOG-PS and mGPS remained significantly associated with CT-derived sarcopenia (CT-SS), the CT-SS was not significantly associated with survival. These results have implications for how CT-derived SMI may be used in the currently proposed GLIM framework to diagnose cancer cachexia.
The limitation of the study consists of that analysis is retrospective in a prospective dataset and may be subject to sample bias.
Author Response
Point taken. The authors thank the reviewer for their feedback.
Reviewer 2 Report
This research is impressive, however there are some format issues need to be addressed. I recommend the acceptance of this manuscript on the Cancers after minor revision.
(1) There are repetitive title description “2. Materials and Methods” “2. Patients and Methods” in page 2.
(2) The figure 1 legend should be put in the blow of figure.
Author Response
Points taken. These changes have been made in the revised version of the manuscript.